# A Fusion Positioning Method for Indoor Geomagnetic/Light Intensity/Pedestrian Dead Reckoning Based on Dual-Layer Tent–Atom Search Optimization–Back Propagation

**DOI:** 10.3390/s23187929

**Published:** 2023-09-16

**Authors:** Yuchen Han, Xuexiang Yu, Ping Zhu, Xingxing Xiao, Min Wei, Shicheng Xie

**Affiliations:** 1School of Earth and Environment, Anhui University of Science and Technology, Huainan 232001, China; hyc_ngu181@163.com (Y.H.);; 2Coal Industry Engineering Research Center of Mining Area Environmental and Disaster Cooperative Monitoring, Anhui University of Science and Technology, Huainan 232001, China; 3School of Geomatics, Anhui University of Science and Technology, Huainan 232001, China; 4School of Geomatics and Urban Spatial Informatics, Beijing University of Civil Engineering and Architecture, Beijing 102616, China

**Keywords:** geomagnetic indoor positioning, PDR, light intensity, BKF-WT, tent-ASO-BP

## Abstract

Indoor positioning using smartphones has garnered significant research attention. Geomagnetic and sensor data offer convenient methods for achieving this goal. However, conventional geomagnetic indoor positioning encounters several limitations, including low spatial resolution, poor accuracy, and stability issues. To address these challenges, we propose a fusion positioning approach. This approach integrates geomagnetic data, light intensity measurements, and inertial navigation data, utilizing a hierarchical optimization strategy. We employ a Tent-ASO-BP model that enhances the traditional Back Propagation (BP) algorithm through the integration of chaos mapping and Atom Search Optimization (ASO). In the offline phase, we construct a dual-resolution fingerprint database using Radial Basis Function (RBF) interpolation. This database amalgamates geomagnetic and light intensity data. The fused positioning results are obtained via the first layer of the Tent-ASO-BP model. We add a second Tent-ASO-BP layer and use an improved Pedestrian Dead Reckoning (PDR) method to derive the walking trajectory from smartphone sensors. In PDR, we apply the Biased Kalman Filter–Wavelet Transform (BKF-WT) for optimal heading estimation and set a time threshold to mitigate the effects of false peaks and valleys. The second-layer model combines geomagnetic and light intensity fusion coordinates with PDR coordinates. The experimental results demonstrate that our proposed positioning method not only effectively reduces positioning errors but also improves robustness across different application scenarios.

## 1. Introduction

Accurate positioning services are essential for both indoor and outdoor environments in today’s rapidly growing economy. While GNSS signals can meet outdoor positioning needs, they are not suitable for indoor positioning [1]. Highly precise indoor positioning not only improves daily activities but also indirectly benefits businesses economically. The continuous development and innovation of wireless devices have given rise to various indoor positioning technologies. These technologies, including Wi-Fi, Bluetooth, and geomagnetic and inertial navigation, have been well researched and have shown satisfactory results [2]. However, these technologies have limitations. Bluetooth, Wi-Fi, and UWB significantly enhance positioning accuracy. However, they demand expensive equipment, as well as continuous maintenance and updates, making these options difficult to scale [3]. Conversely, geomagnetic and inertial navigation positioning provides strong continuity and resistance to environmental interference. These advantages can be achieved without the need for additional physical infrastructure, making these technologies promising for future development. Inertial navigation commonly utilizes PDR methods to obtain positioning results. However, this process tends to accumulate errors [4]. Additionally, single geomagnetic positioning techniques are hampered by low resolution and a high likelihood of fingerprint database mismatches, resulting in significant positioning errors [5]. Fusion methods can be employed to combine different positioning technologies, compensating for their limitations while leveraging their strengths. This increases the diversity and universality of indoor positioning applications and further enhances positioning accuracy. 

Presently, there are two principal classifications for single geomagnetic positioning methods: sequence-based magnetic matching positioning (SBMP) and single-point-based magnetic matching positioning (SPMP) [6]. The application of SBMP technology commonly entails the deployment of the Dynamic Time Warping (DTW) algorithm. Referenced studies [7] introduce a classification scheme to distinguish between quasi-static and non-quasi-static ranges, which enables the integration of inertial recurrence information into the geomagnetic DTW framework. This integration demonstrates improved performance in terms of the matching success rate and reliability in position correction. Zhao et al. [8] presented an advanced methodology, known as IMC-GSS, for the construction of indoor maps. This approach harnesses the power of smartphones and crowdsourcing techniques to collect magnetic trajectory data, subsequently employing DTW clustering to generate highly accurate and reliable indoor maps. The significance of SBMP-based studies is widely recognized in current research. Chen et al. [9] deployed magnetic sensor arrays and recursive probabilistic neural networks (RPNN) to bolster the stability and precision of magnetic field fingerprint maps, thereby proficiently detecting magnetic field anomalies and spatial variations. Sarcevic et al. [10] proposed a fingerprint-based indoor positioning method. They used a multilayer perceptron neural network to determine positions by combining the magnetic field strength with RSSI values. Shu et al. [11] conducted a study introducing a novel approach for indoor geomagnetic positioning using a direction-aware multiscale recurrent neural network (DM-RNN). This deep learning method used an integrated learning approach, yielding robust positioning results. Similarly, Ashraf et al. [12] combined a convolutional neural network (CNN) with geomagnetic field patterns for indoor positioning. They constructed a comprehensive database utilizing MP and implemented a voting mechanism that effectively combined multiple CNN predictions. 

The research on single geomagnetic positioning mainly focuses on data feature extraction techniques and the profound optimization of algorithmic matching. Moreover, the fusion of other positioning technologies with geomagnetic positioning holds immense potential for elevating the precision and robustness of positioning outcomes. Huang et al. [13] proposed a multi-level fusion indoor positioning technology that enhances robustness and reliability in complex environments by integrating pseudo-satellites, Wi-Fi, and the geomagnetic field using an unsupervised learning model. Xu et al. [14] proposed an integrated methodology for fusing Wi-Fi and geomagnetic data. The approach involved training multiple classifiers by constructing fingerprints offline and determining global dynamic fusion weights. During the online phase, the fusion process incorporated the K-nearest neighbors (KNN) technique for selecting matching weights. The literature [15] combines the Extended-Kalman Filter (EKF) with magnetic positioning and PDR techniques. Moreover, the authors conducted gait recognition training using a cognitive evolutionary algorithm, alongside a genetic algorithm to facilitate the search for the optimal magnetic position. Hu et al. [16] analyzed the magnetic interference distribution model of the geomagnetic field and extracted the magnetic data using classification and regression tree algorithms combined with a Kalman filter for positioning. The validation results show that the system provides more than a 70% improvement in positioning accuracy. Yang et al. [17] proposed an indoor localization method that integrates 5G, geomagnetism, and Visual Inertial Odometry (VIO). By employing an Error Backpropagation Neural Network (BPNN) model and an Error State Extended Kalman Filter (ES-EKF), the approach effectively enhances the accuracy and reliability of long-term navigation and positioning. Momose et al. [18] proposed an indoor localization algorithm based on a particle filter with floating maps, which integrates distance data from Bluetooth beacons as well as accelerometer and magnetometer sensors to enhance accuracy. Gao et al. [19] employed a hybrid indoor localization method that integrates Bluetooth beacons, geomagnetic fields, Inertial Measurement Units (IMU), and camera sensors to support various types of indoor applications. This approach maintains a localization accuracy within 2 m. Daou et al. [20] employed deep learning and built-in smartphone sensors to merge visual data with magnetic headings, achieving the high-precision indoor localization of users through a single captured image. 

The studies mentioned above offer profound guidance for further research in geomagnetic positioning. We conclude that using only geomagnetic data for high-precision positioning is challenging. This is because it either requires complicated data preprocessing or complex positioning models. Using other indoor signal sources along with geomagnetic data can simplify the computations. However, this approach requires extra physical infrastructure, increasing the system’s overall cost. This paper introduces an innovative method that synergistically combines indoor visible light factors with geomagnetic signals, allowing for accurate and reliable indoor positioning with just a smartphone. Visible light positioning typically employs two methods. The first method is based on the positioning of light signals from specific equipment. This method decodes the information in these signals and calculates feature values for different locations. When combined with positioning algorithms, it enables accurate positioning. This approach has been extensively studied by researchers [21,22]. The second method utilizes the distribution of light intensity from indoor lighting equipment for positioning. Light intensity refers to the visible-light irradiation on the receiving equipment, measured in lux, which represents the intensity of ambient light around the receiving device. Many buildings today are equipped with fixed indoor lighting equipment, and research indicates that the light intensity received is stronger when closer to the light source in the room, showing a proportional relationship [23]. However, light intensity shows an irregular distribution due to factors like walls, glass reflections, and the characteristics of the collection equipment. Despite this, the irregularity provides the feasibility for using ambient light intensity for positioning. Similar to geomagnetic indoor positioning technology, this positioning method also faces the challenge of low resolution.

This paper integrates indoor geomagnetic three-axis data, light intensity data, and inertial sensor positioning results to propose a two-layer Tent-ASO-BP fusion positioning method based on a smartphone platform. The main processes are as follows:(1)To mitigate errors from device attitude changes, we first process sensor data using coordinate transformation. Next, we incorporate indoor lighting factors to improve spatial resolution. Geomagnetic and light intensity data are processed separately using radial basis function interpolation at different resolutions, establishing a combined fingerprint library.(2)To address the limitations of traditional BP neural networks, like slow convergence and overfitting, we employ the Tent-chaos-mapping-enhanced ASO algorithm to optimize model weights and thresholds, constructing our Tent-ASO-BP positioning model.(3)For estimating pedestrian trajectories, we set a time threshold to eliminate the impact of false wave peaks or troughs in step detection. Furthermore, we design a BKF-WT model to enhance the precision of the pedestrian trajectory projection.(4)We introduce a hierarchically optimized method that fuses multiple data sources for positioning. Using a dual-layer Tent-ASO-BP model, we achieve the final results that integrate geomagnetic, light intensity, and PDR data. We then evaluate our method under various scenarios, comparing it against standalone geomagnetic, geomagnetic/light intensity, and PDR positioning approaches. These comparative tests validate the benefits of our proposed positioning method.

## 2. Geomagnetic/Light Intensity/PDR Fusion Positioning Solution

This paper introduces a hierarchically optimized indoor positioning method through information fusion, as depicted in Figure 1. The first layer uses a tightly coupled structure to merge indoor geomagnetic and light intensity data. The second layer employs a loosely coupled structure to integrate the PDR coordinates with the fused positioning coordinates derived from geomagnetic and light intensity data. The specific approach is as follows: First, we construct the fingerprint library for geomagnetic and light intensity data during the offline phase. We then build the first layer of the Tent-ASO-BP fusion positioning model to perform regression prediction for feature-level fusion, obtaining the geomagnetic and light intensity fusion positioning coordinates. Second, we estimate the pedestrian movement trajectory using an optimized PDR method and apply the second layer of the Tent-ASO-BP algorithm for regression prediction of decision-level fusion. We construct a 4D matrix of geomagnetic and light intensity fusion coordinates and PDR positioning coordinates as input, with a 2D matrix of real coordinate values as output. This process ultimately yields the geomagnetic, light intensity, and PDR fusion positioning results.

### 2.1. Improved PDR Method

The fundamental principle of PDR involves determining a pedestrian’s initial position. It then uses an inertial measurement device, carried by the pedestrian, to continuously monitor various movement parameters. These parameters include geographic direction, acceleration, and step count. By integrating these parameters mathematically, we can calculate the displacement for each step. This allows us to derive the user’s relative motion trajectory within a specific area [24], as shown in Figure 2. However, factors like sensor errors in PDR positioning inevitably accumulate errors over time. This leads to significant deviations in the trajectory. Therefore, it becomes necessary to employ other positioning methods to compensate for this effect.

The formula for deriving the PDR positioning coordinates is represented in Equation (1):(1)$$\left\{\begin{array}{l} x_{i}=x_{i-1}+d_{i}\cos n_{i}\\ y_{i}=y_{i-1}+d_{i}\sin n_{i} \end{array}\right.$$
where $x_{i}$ and $y_{i}$ are the positioning coordinates for the *i*-th step, $d_{i}$ represents its step length, and $n_{i}$ denotes the pedestrian heading angle in its geographic coordinate system. Precise positioning via PDR focuses on three primary aspects:(1)Computing the total step count in the motion trajectory to determine motion distance.(2)Estimating the step size during movement.(3)Calculating the pedestrian’s heading angle, which depicts the direction accurately at each step.

The specific computation methods for each aspect are elaborated below. 

#### 2.1.1. Step Count Estimation

In this paper, we employ the peak detection method for step count estimation. Each step’s acceleration trend resembles a sinusoidal function due to the gait characteristics when walking. This function consists of one cycle containing both peaks and valleys [25]. To eliminate false peaks and valleys, we use stride time to filter out peaks or valleys with closely spaced time intervals. Typically, the stride time interval for pedestrians walking at a uniform speed is not less than 0.5 s. If the time interval between adjacent peaks or troughs is less than 0.5 s, they are considered false. In such cases, we remove the peaks or troughs with smaller absolute values. To mitigate errors, we first pre-process the acceleration data. These errors may arise from irregular sensor movements during walking or high-frequency noise from hardware devices. We initially apply a low-pass filter to denoise the triaxial accelerometer data, and we then calculate the combined acceleration magnitude of the three linear axes using Equation (2): (2)$$a=\sqrt{a_{x}{}^{2}+a_{y}{}^{2}+a_{z}{}^{2}}-g$$
where *a* represents the combined acceleration magnitude; $a_{x}$, $a_{y}$, and $a_{z}$ are the acceleration along the *x*, *y*, and *z* axes, respectively, as measured by the accelerometer; and *g* is the local gravitational acceleration. The overall process of step count estimation is as follows: (1)Set the detection threshold based on the low-pass-filtered acceleration signal.(2)Identify and mark the peak positions. If false peaks are present, use the discrimination method mentioned earlier to reject them. Subtract one from the number of peaks and record the result.(3)Identify and mark the trough positions. If false troughs are present, use the discrimination method mentioned earlier to reject them. Subtract one from the number of troughs and record the result.(4)Finally, output the overall number of peaks and troughs. Use their corresponding markers to determine the total number of detected steps.

Figure 3 presents an illustration of the step count estimation results derived from the method proposed in this paper. We conducted several experiments with different stride lengths while maintaining a constant speed. The average accuracy of our stride count estimation exceeded 96%.

#### 2.1.2. Step Length Estimation

When a pedestrian moves at a constant speed, their step length remains relatively consistent. Several researchers have identified a correlation between step length and the peak acceleration during a stride. The researchers proposed a nonlinear model that estimates the step length based on the observed maximum and minimum acceleration values within a stride [26]. Other notable and widely used models include those by Weinberg [27], Scarlet [28], and Kim [29]. These nonlinear models are not constrained by variables such as an individual’s height or stride frequency. The corresponding formulas are provided in Equations (3)–(5):(3)$$d_{k}=K_{}\cdot \sqrt[{4}]{a_{\max}-a_{\min}}$$
(4)$$d_{k}=K_{}\cdot \frac{\frac{\sum_{i=1}^{N}\left|a_{i}\right|}{N}-a_{\min}}{a_{\max}-a_{\min}}$$
(5)$$d_{k}=K_{}\cdot \sqrt[{3}]{\frac{\sum_{i=1}^{N}\left|a_{i}\right|}{N}}$$
where $d_{k}$ denotes the step length of the *k*-th step; $K_{1}$ is a model parameter, which generally needs to be set artificially according to different testers; $a_{\max}$ and $a_{\min}$ denote the maximum and minimum values of acceleration of the *k*-th step, respectively; and $a_{i}$ represents the measured acceleration value of the *i*-th sample in each stride. In this study, we opt for the widely used Weinberg model for step length estimation. To validate the model’s accuracy, we choose test subjects of varying heights and genders. They are instructed to walk at a uniform pace with a step length of 0.6 m. The average step length estimation results obtained from three different models are presented in Figure 4. It is evident that the Weinberg model demonstrates superior accuracy and stability in step length estimation compared to the other two models.

#### 2.1.3. Heading Angle Calculation Based on BKF-WT

In PDR positioning, there are two traditional methods for calculating the heading angle. Method 1 uses gyroscope data to integrate and solve the angular velocity, resulting in the heading angle [30]. Method 2 uses magnetometer XYZ data to determine the angle between the geomagnetic north and the phone, thereby obtaining the heading angle [31]. Among these methods, Method 1 has the advantage of being unaffected by external environmental disturbances and remaining stable over short periods. However, it may introduce offset errors caused by the gyroscope not completing a full rotation. On the other hand, Method 2 does not generate offset errors and remains stable over long periods. However, it is susceptible to disturbances from external magnetic fields, which can result in significant heading deviations at certain positions.

Some scholars have applied the Kalman filter to merge heading angles calculated from both the gyroscope and magnetometer, compensating for their individual shortcomings [32]. In this paper, we build upon this approach by introducing the Biased Kalman Filter–Wavelet Transform (BKF-WT) for fusion. WT is a signal analysis method based on local characteristics of the signal. It decomposes the signal into a series of wavelet coefficients at different scales, with each representing the energy contribution of the signal at the corresponding scale and frequency. This decomposition can be employed to extract local features of the signal, and the wavelet transform has the advantage of being able to handle non-stationary signals. It also possesses the properties of local and multi-scale analysis, which makes it widely used in signal processing [33].

First, the magnetometer and gyroscope data are wavelet-denoised. Then, the data are fused using the BKF, which is calculated as follows: Based on the Kalman filter principle [34], the state transfer equation is defined in this paper as Equation (6):(6)$$X_{k}=AX_{k-1}+q_{k-1}$$
where $X_{k}={\left[\begin{array}{cc} \varphi_{k}^{gyr} & w_{k}^{gyr} \end{array}\right]}^{T}$ is the state vector; $\varphi_{k}^{gyr}$ and $\varphi_{k}^{gyr}$ are the gyroscope heading angle and angular velocity at time *k*, respectively; the state transfer matrix $A={\left[\begin{array}{cccc} 1 & △t_{k} & ; & \begin{array}{cc} 0 & 1 \end{array} \end{array}\right]}^{T}$; $△t_{k}=t_{k}-t_{k-1}$; $t_{k}$ is the time node at time *k*; and $q_{k-1}$ is the process noise.

The measurement equation of the system can be expressed as Equation (7).
(7)$$Z_{k}=BX_{k}+r_{k-1}$$
where $Z_{k}={\left[\begin{array}{cc} \varphi_{k}^{mag} & w_{k}^{mag} \end{array}\right]}^{T}$ represents the observed value at the *k*-th moment, while $\varphi_{k}^{mag}$ and $w_{k}^{mag}$ denote the magnetometer heading angle and the time derivative of the magnetometer heading angle at the *k*-th moment, respectively. *B* is the unit matrix and $r_{k-1}$ symbolizes the observation noise.

The Kalman filter operates in two stages: prediction and update. The prediction equation is given by Equation (8).
(8)$$\left\{\begin{array}{l} X_{k|k-1}^{-}=A_{k|k-1}X_{k-1|k-1}\\ P_{k|k-1}^{-}=A_{k|k-1}P_{k-1|k-1}A_{k|k-1}+Q \end{array}\right.$$
where $X_{k|k-1}^{-}$ is the a priori state estimation matrix; $P_{k|k-1}^{-}$ is the a priori covariance matrix; and Q is the process noise matrix. The update equation is given by Equation (9):(9)$$\left\{\begin{array}{l} K_{k}=P_{k|k-1}^{-}B^{T}{\left[BP_{k|k-1}^{-}B^{T}+R\right]}^{-1}\\ X_{k|k}=X_{k|k-1}^{-}+K_{k}\left(Z_{k}-BX_{k|k-1}^{-}\right)\\ P_{k|k}^{}=(I-K_{k}B)P_{k-1|k-1} \end{array}\right.$$
where $K_{k}$ is the Kalman gain; $X_{k|k}$ is the posterior state matrix; $P_{k|k}^{}$ is the posterior covariance matrix; *R* is the observation noise matrix; and *I* is the unit array. Subsequently, the idea of biased estimation is applied to obtain a biased Kalman filter that is superior to the conventional KF by biasing the results of the Kalman filter. It is defined as Equation (10):(10)$${\overset{⌢}{X}}_{k}^{BKF}=\gamma_{k}{\overset{⌢}{X}}_{k}^{KF}$$
where ${\overset{⌢}{X}}_{k}^{BKF}$ is the BKF estimate, $\gamma_{k}$ is the bias parameter, and ${\overset{⌢}{X}}_{k}^{KF}$ is the KF estimate. Then, the deviation and variance of ${\overset{⌢}{X}}_{k}^{BKF}$ can be expressed as Equations (11) and (12), respectively:(11)$$\mathrm{bias}({\overset{⌢}{X}}_{k}^{BKF})=E[{\overset{⌢}{X}}_{k}^{BKF}-X_{k}]=E[\gamma_{k}{\overset{⌢}{X}}_{k}^{KF}-X_{k}]=(\gamma_{k}-1)X_{k}$$
(12)$$\mathrm{var}({\overset{⌢}{X}}_{k}^{BKF})=\mathrm{var}[\gamma_{k}{\overset{⌢}{X}}_{k}^{KF}]=\gamma_{k}{}^{2}Tr(P_{k|k})$$

Then, the mean squared error of ${\overset{⌢}{X}}_{k}^{\mathrm{BKF}}$ can be expressed as Equation (13):(13)$$MSE({\overset{⌢}{X}}_{k}^{\mathrm{BKF}})=\gamma_{k}{}^{2}Tr(P_{k|k})+{\left(\gamma_{k}-1\right)}^{2}{\left\|X_{k}\right\|}^{2}$$

To minimize $MSE({\overset{⌢}{X}}_{k}^{\mathrm{BKF}})$, which is equivalent to the problem of finding the minimum value of $MSE({\overset{⌢}{X}}_{k}^{\mathrm{BKF}})$, the derivative of $MSE({\overset{⌢}{X}}_{k}^{\mathrm{BKF}})$ with respect to *k* is taken and equated to zero, which results in Equation (14).
(14)$$\gamma_{k}=\frac{{\left\|X_{k}\right\|}^{2}}{Tr(P_{k|k})+{\left\|X_{k}\right\|}^{2}}$$
where ${\left\|\;•\;\right\|}^{2}$ denotes the Euclidean parametrization of the vector; then, the complete equation of the biased Kalman filter can be expressed as Equation (15):(15)$$\left\{\begin{array}{l} X_{k|k-1}^{-}=A_{k|k-1}X_{k-1|k-1}\\ P_{k|k-1}^{-}=A_{k|k-1}P_{k-1|k-1}A_{k|k-1}+Q\\ K_{k}=P_{k|k-1}^{-}B^{T}{\left[BP_{k|k-1}^{-}B^{T}+R\right]}^{-1}\\ P_{k|k}^{}=(I-K_{k}B)P_{k-1|k-1}\\ {\overset{⌢}{X}}_{k}^{KF}=X_{k|k-1}^{-}+K_{k}\left(Z_{k}-BX_{k|k-1}^{-}\right)\\ \gamma_{k}=\frac{{\left\|X_{k}\right\|}^{2}}{Tr(P_{k|k})+{\left\|X_{k}\right\|}^{2}}\\ {\overset{⌢}{X}}_{k}^{BKF}=\gamma_{k}{\overset{⌢}{X}}_{k}^{KF}\\ P_{k}^{BKF}=\gamma_{k}{}^{2}P_{k|k}\\ X_{k|k}={\overset{⌢}{X}}_{k}^{BKF}\\ P_{k|k}=P_{k}^{BKF} \end{array}\right.$$

### 2.2. Geomagnetic/Light Intensity Fusion Positioning

#### 2.2.1. Coordinate System Conversion

The built-in sensor of a smartphone can collect the magnetic field intensity of the geomagnetic XYZ three-axis direction anywhere in a room. However, the data are under the carrier coordinate system, and the phone’s posture will inevitably change during the walking process, causing the collected geomagnetic XYZ three-axis data to change as well. If the collected initial geomagnetic data are not processed, large errors may occur and affect the positioning performance. To avoid this, a database is commonly built using the modulus of the triaxial component of the magnetic field signal [35]. The calculation formula for the magnetic field signal modulus is given in Equation (16):(16)$$M=\sqrt{M_{x}^{2}+M_{y}^{2}+M_{z}^{2}}$$
where M indicates the modulus of triaxial magnetic field signals; and $M_{x}$, $M_{y}$, and $M_{z}$ represent the triaxial data collected by the smartphone. However, this method reduces the resolution of the geomagnetic fingerprint database, which may result in a decrease in positioning performance. Therefore, the geomagnetic data collected under the carrier coordinate system can be converted to the more stable geomagnetic triaxial data under the geographic coordinate system. This is performed using coordinate system conversion methods. The conversion process is as follows:

Let $M_{x}^{\gamma }$, $M_{y}^{\gamma }$, and $M_{z}^{\gamma }$, respectively, denote the converted geographic triaxial data. The orientation changes of the smartphone during data acquisition include the pitch (φ), heading (γ), and roll angle (θ). These angles can be directly obtained from the sensors integrated into the smartphone. The conversion equations are shown in Equation (17):(17)$$C_{x}=\left[\begin{array}{ccc} 1 & 0 & 0\\ 0 & \cos\varphi & -\sin\varphi \\ 0 & \sin\varphi & \cos\varphi \end{array}\right]\;C_{y}=\left[\begin{array}{ccc} \cos\theta & 0 & \sin\theta \\ 0 & 1 & 0\\ -\sin\varphi & 0 & \cos\varphi \end{array}\right]\;C_{z}=\left[\begin{array}{ccc} \cos\gamma & -\sin\gamma & 0\\ \sin\gamma & \cos\gamma & 0\\ 0 & 0 & 1 \end{array}\right]$$
where $C_{x}$, $C_{y}$, and $C_{z}$ represent the rotation matrix corresponding to the pitch angle, heading angle, and roll angle, respectively. The expression of the final rotation matrix is shown in Equation (18):(18)$$C_{n}=C_{z}C_{y}C_{x}=\left[\begin{array}{ccc} \cos\varphi \cos\gamma & -\cos\theta \sin\gamma +\sin\theta \sin\varphi \cos\varphi & \sin\theta \sin\gamma +\cos\theta \sin\varphi \cos\gamma \\ \cos\varphi \sin\gamma & \cos\theta \cos\gamma +\sin\theta \sin\varphi \sin\gamma & -\sin\theta \cos\gamma +\cos\theta \sin\varphi \sin\gamma \\ -\sin\varphi & \sin\theta \cos\varphi & \cos\theta \cos\varphi \end{array}\right]$$

The conversion equation is shown in Equation (19).
(19)$$\left(\begin{array}{l} M_{x}^{\gamma }\\ M_{y}^{\gamma }\\ M_{z}^{\gamma } \end{array}\right)=C_{n}\left(\begin{array}{l} M_{x}\\ M_{y}\\ M_{z} \end{array}\right)$$

#### 2.2.2. Spatial Interpolation

The number of fingerprints in the geomagnetic fingerprint database is an important factor affecting its resolution, as too few fingerprints can lead to an increased probability of false matches. Therefore, spatial interpolation can be employed to interpolate the initially collected fingerprint library to obtain a sufficient number of fingerprints and improve its resolution while ensuring a constant workload [36]. Radial Basis Function (RBF) interpolation is a more accurate method that, given a sufficient amount of data, can fit the best smooth surface and is more efficient. Thus, this paper chooses the RBF interpolation method for interpolating different resolutions. Taking the geomagnetic X-axis data and light intensity data in a section of a corridor as an example, the interpolation effect is shown in Figure 5.

## 3. Tent-ASO-BP Model

### 3.1. Backpropagation Neural Network

Artificial Neural Networks (ANNs) have emerged as a research hotspot in machine learning models in recent years, finding extensive use in function prediction, pattern recognition, and other fields [37]. The backpropagation (BP) algorithm is a supervised learning algorithm used in ANNs. It trains the network by receiving sample signals. These signals pass through the input layer and one or more hidden layers to reach the output layer. Typically, the output result falls short of expectations, prompting the BP neural network to feed back the error. The hidden layers then continuously adjust their “state” based on the error information, allowing the improved signal to be re-processed for the resulting output. If the result still fails to meet the requirements, the process is repeated until a satisfactory output is obtained [38]. Figure 6 displays the structure of the BP neural network for the geomagnetic and light intensity tightly coupled layer, as constructed in this paper.

The number of neurons in the hidden layer of the BP neural network is shown in Equation (20):(20)$$g=\sqrt{h+l}+a$$
where *h* and *l* are the numbers of nodes in the input and output layers, respectively, and *a* is a constant in the range of [1, 10]. Its mean square error function is shown in Equation (21):(21)$$E=\frac{1}{2}\sum_{n=1}^{i}{\left(y_{h}-\overset{\wedge }{y_{h}}\right)}^{2}$$
where *i* is the number of neurons in the output layer, $y_{h}$ is the desired output, and $\overset{\wedge }{y_{h}}$ is the output of the output layer after the *h*-th iteration.

### 3.2. ASO Algorithm

Atom Search Optimization (ASO) is a population intelligence optimization algorithm developed based on the physical phenomenon of atomic motion in molecular dynamical systems [39]. Within a specific molecular system, atoms are subject to interaction forces, which prompt them to perform an overall spatial search due to their geometric constraints and internal motions. The gravitational force causes less massive atoms to move closer to more massive ones more quickly, enabling them to reach optimal spatial positions. Meanwhile, the repulsive force disperses the atoms into various regions, expanding the overall space and developing more possible regions for optimal solutions. 

In the *n*-dimensional space consisting of *s* atoms, the potential energy received between atoms changes as the number of iterations changes. The mutual potential energy received by a particular atom *i* after the *q*-th iteration is shown in Equation (22):(22)$$\left\{\begin{array}{l} F_{ij}^{n}(q)=\lambda (q)\left[2{\left(h_{ij}(q)\right)}^{3}-{\left(h_{ij}(q)\right)}^{7}\right]\\ \lambda (q)=\beta {\left(1-\frac{q-1}{Q}\right)}^{3}e^{-\left(\frac{20q}{Q}\right)} \end{array}\right.$$
where $F_{ij}(q)$ denotes the mutual potential energy of atoms *i*, *j* after the *q*-th iteration, β is the depth weighting factor, *Q* represents the total number of iterations, and $h_{ij}(q)$ denotes the nature of the force that varies with the distance of the atoms. The expressions are shown in Equation (23):(23)$$h_{ij}=\left\{\begin{array}{l} h_{\min}=\varepsilon +\varepsilon (q),\frac{r_{ij}}{\sigma (q)}<h_{\min}\\ \frac{r_{ij}}{\sigma (q)},h_{\min}\leq \frac{r_{ij}}{\sigma (q)}\leq h_{\max}\\ h_{\max},\frac{r_{ij}}{\sigma (q)}>h_{\max} \end{array}\right.$$
where $h_{\max}$ and $h_{\min}$ denote its lower and upper bounds, respectively; and ε is a shift factor that makes the atoms move better when the number of iterations increases to search the global and exploit unknown regions. $r_{ij}$ denotes the Euclidean distance between atoms *i* and *j*, and σ(q) denotes the collision diameter between atoms.

From the above, the potential energy exerted by atom *i* by atom *j* is shown in Equation (24):(24)$$F_{i}^{n}(q)=\sum_{j\in TBest}rand_{j}F_{ij}^{n}(q)$$
where $rand_{j}$ is a random number in the range of [0~1]; *T* denotes the set of atoms other than atom *i* that have better adaptation performance, and the expression is $T=s-(s-2)\sqrt{\frac{q}{Q}}$; and *T-Best* denotes the optimal solution in this set.

Additionally, the mutual binding force plays a crucial role in illustrating the motion law between atoms. In an *n*-dimensional space, each atom is influenced by the binding force from the atom in the best position, and this force changes with the number of iterations. The expression for this is given in Equation (25):(25)$$\left\{\begin{array}{l} G_{i}^{n}(q)=\mu (q)\left(x_{Best}^{n}(q)-x_{i}^{n}(q)\right)\\ \mu (q)=\alpha e^{-\left(\frac{20q}{Q}\right)} \end{array}\right.$$
where μ(q) denotes the adaptive coefficient term with the number of iterations, α denotes the weight coefficient, and $x_{Best}^{n}(q)$ denotes the best spatial position obtained after *q* iterations.

The atomic masses change with the number of iterations. In an *n*-dimensional space consisting of *s* atoms, the estimated mass of atom *i* after *q* iterations is shown in Equation (26):(26)$$M_{i}(q)=e^{-\left(\frac{{Fit}_{i}(q)-{Fit}_{1}(q)}{{Fit}_{2}(q)-{Fit}_{1}(q)}\right)}$$
where $Fit_{i}(q)$ denotes the value of the objective function of atom *i* after the *q*-th iteration, and the subscripts 1 and 2 represent the maximum and minimum values of this objective function, respectively. Then, the mass of the atom is shown in Equation (27):(27)$$m_{i}(q)=\frac{M_{i}(q)}{\sum_{j=1}^{s}M_{j}(q)}$$

Since the motion of an atom follows the laws of physics, it follows from Newton’s second law that the acceleration of an atom is inversely proportional to its mass and varies with the number of iterations. The expression for the acceleration of atom *i* is shown in Equation (28):(28)$$a_{i}^{n}(q)=\frac{F_{i}^{n}+G_{i}^{n}(q)}{m_{i}^{n}(q)}$$

In summary, the expression for the velocity and position update of this atom in *n*-dimensional space is shown in Equation (29):(29)$$\begin{array}{l} v_{i}^{n}(q+1)=rand_{i}^{n}v_{i}^{n}(q)+a_{i}^{n}(q)\\ p_{i}^{d}(q+1)=p_{i}^{d}(q)+p_{i}^{d}(q+1) \end{array}$$

While the ASO demonstrates excellent optimization performance and has seen widespread use in recent years, it does have limitations. Specifically, the algorithm tends to converge to local optima. Therefore, there is room for improvement in its search efficiency.

### 3.3. Tent Chaotic Map

Chaos is characterized by the random, irregular movement of individuals within a deterministic system, resulting in non-repetitive, uncertain, and unpredictable behavior. In ASO, these features can be utilized to maintain global search ability and population diversity more effectively, avoiding local optima. Tent chaotic mapping has been shown to have a faster search speed and better traversal uniformity compared to other chaotic mappings [40]. The expression for Tent chaotic mapping is presented in Equation (30):(30)$$x_{k+1}=\left\{\begin{array}{l} 2x_{k},0\leq x\leq \frac{1}{2}\\ 2(1-x_{k}),\frac{1}{2}<x\leq 1 \end{array}\right.$$

The process is as follows: A constant within the range of (0, 1) is randomly chosen as the initial value $x_{0}$. Subsequent iterations are then computed according to the aforementioned equation, continuously generating chaotic sequences as the iterations progress. The process halts when the iteration count *k* reaches its maximum value. All generated chaotic sequences are then saved.

### 3.4. Tent-ASO-BP Positioning Model

The core of the position model proposed in this paper lies in determining the optimal position of the atom by the merit-seeking nature of the Tent-ASO algorithm. The position information corresponds to the initial weights and thresholds of the BP neural network. The Tent-ASO algorithm is used to solve it cyclically until the optimal combination of parameters is obtained. The optimized BP neural network model is then used for coordinate prediction. The algorithm flow is as follows:Step 1:Initially, configure the neural network parameters: a maximum training epoch of 1000, a default learning rate of 0.01, and a momentum factor of 0.5. Subsequently, initialize the atom population size and set the hyperparameter range for each atom, specifying an initial population of 30, a maximum iteration count of 500, and a search dimensionality of 30.Step 2:Generate chaotic sequences using Tent mapping to regenerate the initialized atomic population positions.Step 3:Determine the fitness function of the atom and calculate it to obtain the current optimal position of the atom and the optimal solution. In this paper, the positioning accuracy is used as the fitness function, and the expression is shown in Equation (31):(31)$$G(x)=\sum_{i=1}^{N}\sqrt{{\left({x^{\prime }}_{i}-x_{i}\right)}^{2}+{\left({y^{\prime }}_{i}-y_{i}\right)}^{2}}$$
where *N* is the number of sample points in the database; ${x^{\prime }}_{i}$ and $y_{i}$ are the coordinates of the *i*-th sample prediction locus; and $x_{i}$ and $y_{i}$ are their actual coordinates, respectively.Step 4:Calculate the individual acceleration, velocity, and position update of the atom according to Equations (28) and (29).Step 5:After updating, calculate the fitness of the atomic individuals again to obtain the optimal position and the optimal solution.Step 6:Set the iteration termination criteria. If the optimal fitness is achieved or the maximum number of iterations is reached, terminate the iteration and output the optimal position of the atomic individual. If not, repeat Steps 2–5.Step 7:Execute recurrent prediction in the BP neural network using the weight thresholds optimized in the previous steps to obtain the best performing position model.Step 8:Use input sample sets and test sets to evaluate its position performance. Derive final position results and position errors after inverse normalization of the data. The flow chart of the algorithm is depicted in Figure 7.

## 4. Experiments and Results

### 4.1. Experimental Environment

To minimize randomness in the application of our fusion position method, we consider two experimental scenarios. Experimental Scenario I selects a basement located on the ground floor of a school building as the designated test site. The experiments are carried out during nighttime to minimize the potential interference from sunlight. To validate the accuracy of the results, we set up a relative coordinate system in the test site. The origin is at the southwest corner with coordinates (1,1). The X-axis extends in the eastward direction while the Y-axis runs northward; this coordinate system is also applied to PDR positioning. The dimensions of the test area span 12 m along the X-axis and 26.4 m along the Y-axis. We place markers at 0.3 m intervals along both the X and Y axes. These markers provide relative coordinates as position labels. Notably, the Y-axis is equipped with four identical types of lighting equipment. For a comprehensive understanding of the specific layout, please refer to Figure 8.

Experimental Scenario II investigates how well our geomagnetic, light intensity, and PDR fusion positioning method addresses the common “trajectory through the wall” problem within specific scenarios. For this, we choose a specific corridor section on the school’s fifth floor as our test site. Like in Scenario I, we conduct the experiments at night to avoid sunlight interference. We also independently establish a relative coordinate system at the test site, with the origin point situated in the northeast corner of the test site and designated as (1,1). The positive direction of the X-axis extends westward, while the positive direction of the Y-axis extends southward. We also use this coordinate system for PDR positioning. The corridor has 14 identical lights along the X-axis and 7 along the Y-axis. We place markers every 0.6 m along both axes, dividing the site and providing relative coordinates for each marker. The corridor is 2.4 m wide, 54 m long along the X-axis, and 30 m along the Y-axis. The corridor features 14 identical lighting fixtures positioned strategically along the X-axis direction, complemented by 7 lighting fixtures of the same type along the Y-axis direction. For a detailed layout, see Figure 9.

### 4.2. Geomagnetic/Light Intensity Fingerprint Library Construction

We divided each test area into a 1.2 m × 1.2 m grid and collected geomagnetic and light intensity data at each grid point during the offline phase. The equipment, software, and associated parameters remained consistent across both test sites. Data were collected using a VIVO S9 device and custom-developed physical measurement software. The data collection frequency was set to 50 Hz, and data were collected at each point for 30 s. After collecting the data, we processed the geomagnetic and light intensity data from both scenes using the interpolation method outlined in Section 2.2.2 to create a fingerprint library. Figure 10 and Figure 11 show the distribution of geomagnetic and light intensity data for both scenarios.

### 4.3. Accuracy Experiment of Heading Angle

The experiment, conducted in a corridor, followed the equipment and data acquisition methods outlined in Section 2.1. Here, subjects followed a predetermined path: they walked west at a constant speed until reaching a corner, and then turned 90° to walk north. Finally, they made another 90° turn to walk east. During the experiment, the participant held the smartphone in hand, trying to keep the device horizontally oriented and stable. In this paper, we introduce a metric: the pedestrian single-step heading solution error. This metric measures the accuracy of the model’s heading angle solution. This error can be quantified using Equation (32):(32)$$E_{h}=\frac{1}{N}\sum_{i=1}^{N}\left((H_{e}^{i}-H_{t}^{i})\mathrm{mod}360\right)$$
where $H_{e}^{i}$ and $H_{t}^{i}$ denote the heading solution value and the heading truth value of step-*i*, respectively.

Following data acquisition, WT processing is initially applied to the raw data. Next, the magnetometer heading angle is calculated using the noise-reduced magnetometer data. Then, we use the complex trapezoidal product method to determine the gyroscope heading angle from the noise-reduced gyroscope data. Lastly, the two heading angles are fused using the BKF. The fusion results can be observed in Figure 12 and Figure 13 and Table 1.

Figure 12 and Figure 13 and Table 1 present the heading angle results obtained using different methods. Following wavelet noise reduction, the mean and variance of the error after KF fusion are reduced. Using BKF fusion, the mean and variance of the single-step heading error reach 5.45° and 15.42, respectively. This represents a 5.4% decrease in the mean error and an 18.11% decrease in the variance compared to conventional KF fusion. Employing the BKF-WT algorithm results in 90% of the error distribution being less than 11.37°, which is 29.51% better than the traditional gyroscope heading angle. In summary, the BKF-WT model in this paper enhances the stability of the heading angle solution using smartphone sensors. It also effectively reduces the heading angle deviation during walking.

### 4.4. Performance Experiments of the Position Algorithm

Eighty test points were randomly selected from the geomagnetic/light intensity fingerprint library in scenario I and scenario II. These test points were input into three different algorithms: the traditional BP, ASO-BP, and Tent-ASO-BP. We aimed to compare their respective position errors. The iterative error variations in the above three algorithms are shown in Figure 14.

As shown in Figure 14, the traditional BP neural network has a slow convergence rate. It takes 114 training sessions to bring the Mean Squared Error (MSE) below 0.1. The minimum MSE value achieved is 0.0671. The ASO-BP neural network requires only 66 training sessions to reduce the MSE to below 0.1, with a minimum value of 0.0086. The Tent-ASO-BP neural network takes 48 training sessions to reduce the MSE to below 0.1, with a minimum value of 0.0024. In summary, the ASO-BP algorithm significantly speeds up convergence. Additionally, the BP neural network, when optimized by the Tent-ASO algorithm, converges even faster and minimizes iteration errors, effectively avoiding local optima. The positioning results are shown in Figure 15 and Figure 16.

Figure 15 shows the positioning error for each individual point, while Figure 16 presents the cumulative error distribution for all 80 points. As shown in Figure 16, the traditional BP algorithm has positioning errors mainly within 4 m. In contrast, errors for the ASO-BP and Tent-ASO-BP algorithms are primarily within 3 m and 2 m. Among them, the percentage of precise positioning error within 1 m is 21.2%, 33.7%, and 45%, respectively. 

The corresponding average positioning error is shown in Table 2. Compared to the traditional BP algorithm, the ASO-BP algorithm improves the positioning accuracy by 35% and reduces the standard deviation by 43%. Similarly, the Tent-ASO-BP algorithm increases the accuracy by 47.8% and decreases the standard deviation by 53.4%. The experimental results show that the Tent-ASO-BP positioning model proposed in this paper exhibits both accuracy and stability, making it an effective solution for indoor positioning.

### 4.5. Trajectory Position Experiments

To authenticate the positioning accuracy of the fusion model proposed herein, we carried out pedestrian experiments separately at the experimental sites depicted in Figure 8 and Figure 9. To ensure consistency, the same data acquisition equipment, software, relevant parameter settings, and personnel were deployed for both experimental scenarios. In Experimental Scenario I, we followed the path from point A to D, as shown in Figure 8, covering a total of 43.2 m. In Experimental Scenario II, we started at the START point and followed the arrow in Figure 9, covering a total distance of 80.6 m.

During the experiments, we walked at a uniform speed along the planned route, maintaining a step length of 0.6 m. During the experiment, we held the smartphone in hand, trying to keep the device horizontally oriented and stable, and collected acceleration, geomagnetic triaxial data, and light intensity data in real-time. After data collection, we compared the different position results. The trajectory positioning results for Experimental Scenario I are shown in Figure 17 and Figure 18.

The analysis of Figure 17 reveals that single geomagnetic positioning exhibits a certain degree of the “positioning jump-back” phenomenon. However, fusing it with indoor light intensity data effectively mitigates this issue. Some points, however, still show large deviations. The PDR positioning trajectory closely follows the actual trajectory during the initial stage of positioning, but the deviation at the end of the trajectory increases due to error accumulation. Overall, the positioning trajectory resulting from the fusion of geomagnetic, light intensity, and PDR data is significantly better than the other three positioning methods.

As shown in Figure 18, the trajectory positioning error for the fusion of geomagnetic, light intensity, and PDR data never exceeds 1.25 m. Impressively, 86.4% of positioning instances fall within an accurate range of 1 m. In contrast, the standalone geomagnetic positioning offers an accuracy of merely 45.5%, while the fusion of geomagnetic and light intensity data improves accuracy to 58.1%. Notably, PDR positioning lags behind with an accuracy of only 22.7%. A comprehensive comparison of their positioning errors is presented in Table 3.

The comparative analysis reveals that the geomagnetic/light intensity/PDR fusion positioning method enhances accuracy. When compared to the first three positioning techniques, the method improves accuracy by 0.92 m, 0.42 m, and 1.03 m, respectively. Notably, this fusion method boasts the smallest overall standard deviation, underscoring its superior stability. Thus, the fusion positioning method proposed in this study shows optimal performance in the Scene I experiment. The trajectory positioning outcomes from experimental scenario I are visually represented in Figure 19 and Figure 20:

Figure 19 shows that expanding the positioning area and fingerprint database leads to more mismatches in single geomagnetic positioning. This problem is particularly evident in the corners, causing major errors such as “trajectory through the wall.” Merging geomagnetic and light intensity data lessens mismatches but does not eliminate “positioning jump-back”, particularly in corners prone to errors. PDR positioning starts off aligned with the actual path, but cumulative errors cause it to gradually deviate outside the building. In contrast, our proposed method exhibits remarkable advantages. Our method eliminates errors like “trajectory through the wall” and “positioning jump-back”, closely matching the real trajectory and delivering the most accurate results.

As Figure 20 shows, the fusion approach keeps 83.5% of positioning errors within 1 m, and notably, never exceeds a 2 m error. With standalone geomagnetic data, 72.7% of errors remain within 3 m, but some can exceed 7 m. The fusion of geomagnetic and light data keeps 70.2% of errors within 2 m, but some outliers do exceed 5 m. With PDR-derived errors, 76.1% remain within 3 m, though the maximum can exceed 5 m. The specific position errors of the Scene 2 experiments are presented in Table 4.

The analysis in Table 4 reveals that the fusion position method proposed here outperforms the three alternative methods. Notably, the fusion method excels in key metrics like maximum error, average error, and standard deviation. These findings highlight the method’s high accuracy, reducing errors by 1.8 m, 1.02 m, and 1.66 m. Overall, results from both Scenarios I and II confirm the fusion model’s superior positioning performance.

## 5. Discussion and Conclusions

This paper focuses on indoor positioning using relevant sensors in smartphones and proposes a fused indoor position method based on dual-layer Tent-ASO-BP. First, geomagnetic fingerprints are extracted using the coordinate conversion method. Additionally, the fingerprint resolution is improved by utilizing the characteristics of indoor light intensity and the spatial interpolation method. This increases the correct matching rate while reducing the workload of preliminary fingerprint collection. The geomagnetic/light intensity position results are then output through the first layer of the Tent-ASO-BP model. Subsequently, the PDR method is optimized for step estimation and heading angle estimation, and the final output of geomagnetic/light intensity/PDR position results are obtained through the second layer of the Tent-ASO-BP model. In this paper, experimental validation is carried out in basement and corridor environments. The final average positioning errors are 0.56 m and 0.75 m, which significantly improves the accuracy and stability of the single positioning technique. The conclusions of this paper are as follows:(1)The fusion positioning model presented in this paper is highly convenient. It relies solely on a smartphone to achieve precise and reliable positioning. Significantly, this model mitigates issues like “positioning jump-back” and “trajectory through the wall.” These issues often occur in trajectory-based positioning. As a result, it provides valuable insights and practical guidance for positioning applications in stable lighting environments such as mines, underground parking lots, and tunnels. Even in scenarios without a stable light source, our well-established geomagnetic fingerprint library comes into play. It enables tightly coupled geomagnetic and PDR positioning, enhancing the method’s versatility and robustness. It is important to acknowledge that our study has certain limitations. Specifically, the experimental sites chosen for this research are selected to address more common challenges in indoor positioning. The computational complexity and applicability of the proposed positioning method for larger spaces like shopping malls or areas with special building materials still require further discussion.(2)The dual-layer Tent-ASO-BP model constructed in this study offers several advantages. First, by integrating chaos mapping and intelligent optimization algorithms into the conventional BP neural network’s optimization process, the model significantly accelerates the convergence speed of the BP neural network. This results in improved learning rates and training efficiency. Consequently, the model’s generalization capabilities are substantially enhanced, making it highly effective in dealing with new data instances. Second, the hierarchical structure allows for more efficient processing of different types of information, thereby enhancing the accuracy of real-time positioning and increasing the fault tolerance of the positioning system. Furthermore, this layered architecture easily accommodates new data sources or algorithms. Given that different environments may influence the accuracy of individual information sources, the hierarchical optimization strategy, through the fusion of multiple information sources, serves to overcome such limitations, thereby enhancing the model’s universal applicability. Importantly, this optimization methodology exhibits promise for application across various disciplines and domains.(3)In our experimental setup, we ensure that the smartphone is positioned as flat as possible. However, it is crucial to acknowledge that this approach has inherent limitations. For instance, when pedestrians engage in activities like calling or swinging their phones while walking, the acquired data can be skewed. This is due to constantly changing posture angles, which can significantly influence both acceleration and gyroscope data. As a result, substantial deviations in the heading angle and step count may occur. To address this challenge, our future research endeavors will prioritize the development of sophisticated walking state recognition techniques. Additionally, the novelty of our approach lies in leveraging indoor ambient illumination characteristics for positioning. Unlike conventional methods that rely on visible light signals emitted by LEDs, our approach is independent of lighting equipment and distribution factors. This uniqueness makes our approach relatively innovative and environmentally friendly. We will further explore and investigate its potential.

## Figures and Tables

**Figure 1 sensors-23-07929-f001:**
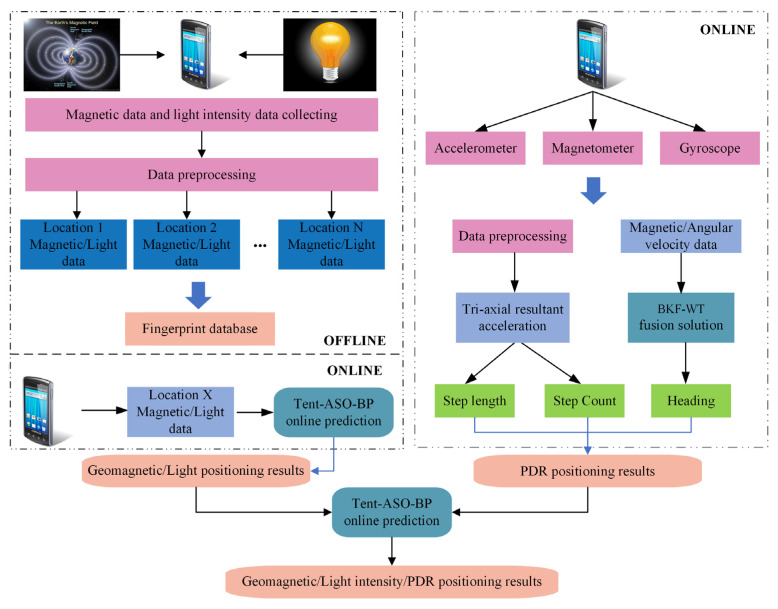
Geomagnetic/light intensity/PDR fusion positioning process.

**Figure 2 sensors-23-07929-f002:**
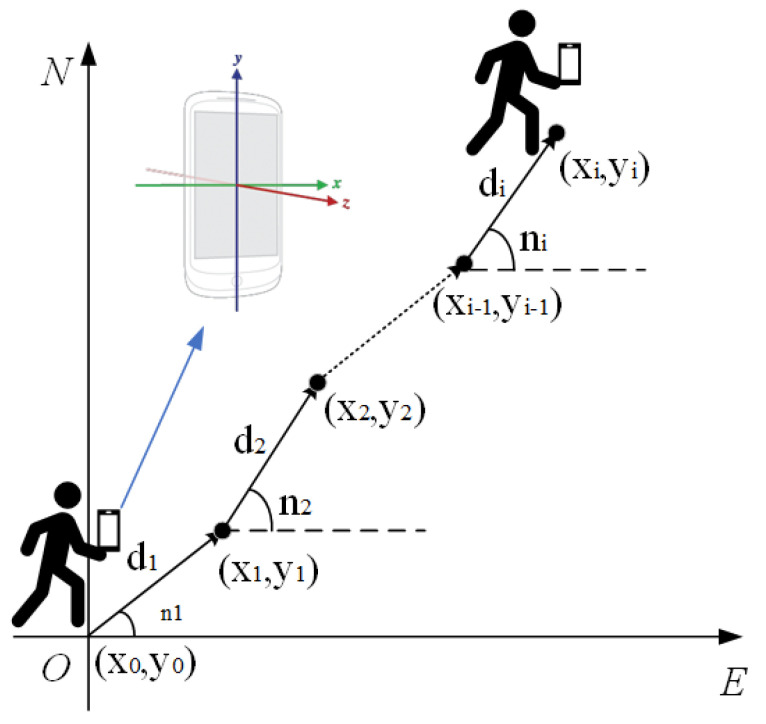
PDR positioning principle diagram.

**Figure 3 sensors-23-07929-f003:**
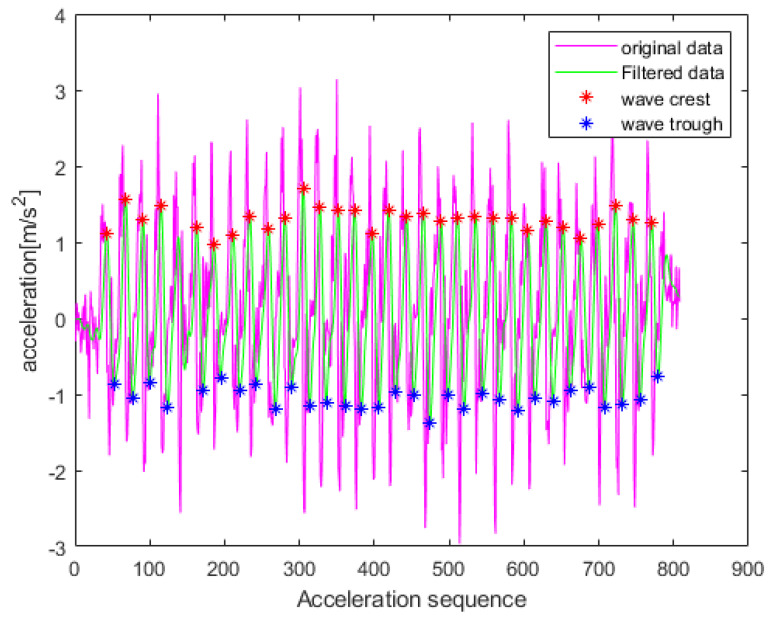
Step count estimation results.

**Figure 4 sensors-23-07929-f004:**
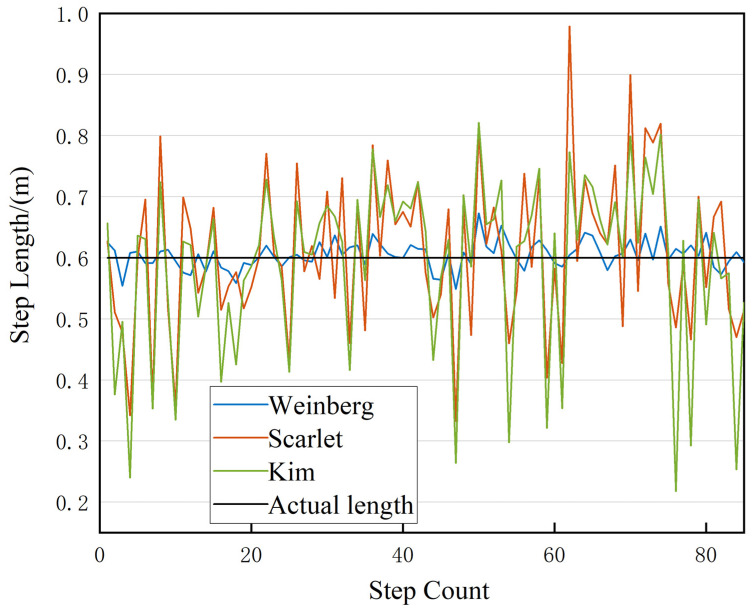
Step length estimation results.

**Figure 5 sensors-23-07929-f005:**
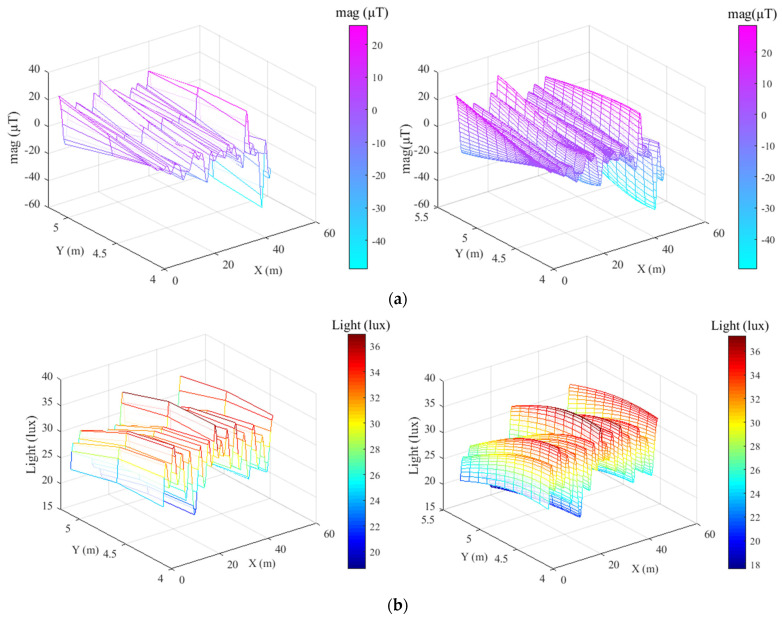
(**a**) Interpolation of geomagnetic data at different resolutions; (**b**) interpolation of light intensity data at different resolutions.

**Figure 6 sensors-23-07929-f006:**
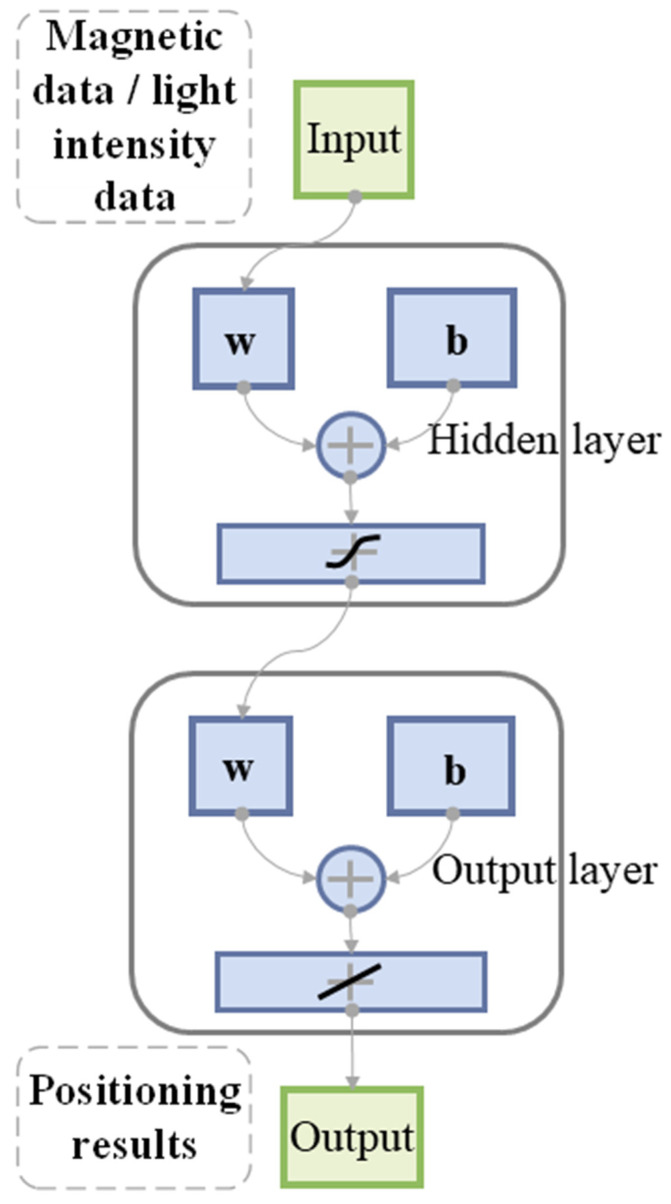
BP neural network architecture diagram.

**Figure 7 sensors-23-07929-f007:**
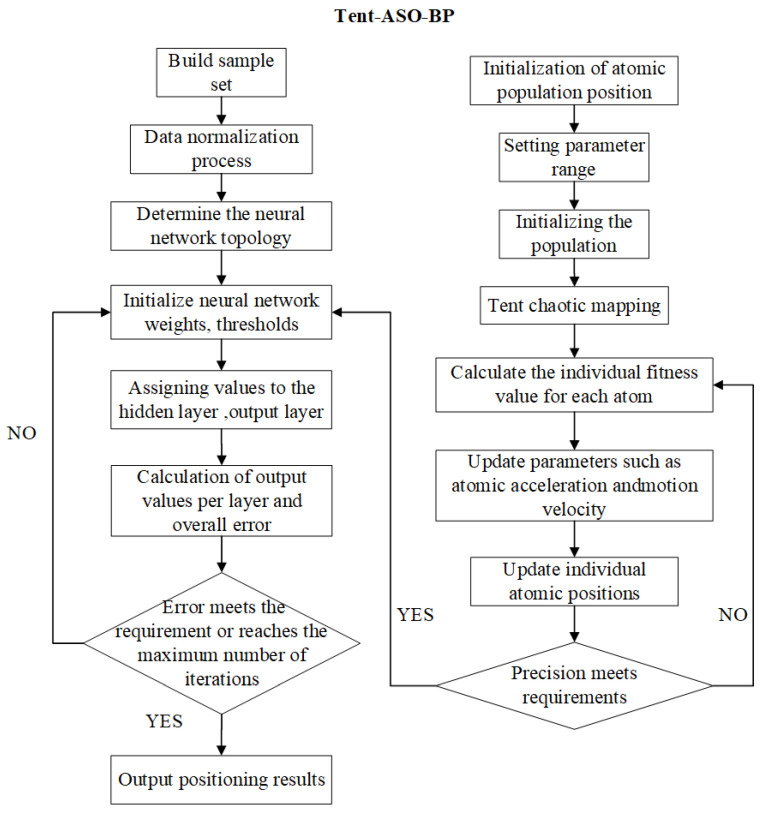
The Tent-ASO-BP model construction flow chart.

**Figure 8 sensors-23-07929-f008:**
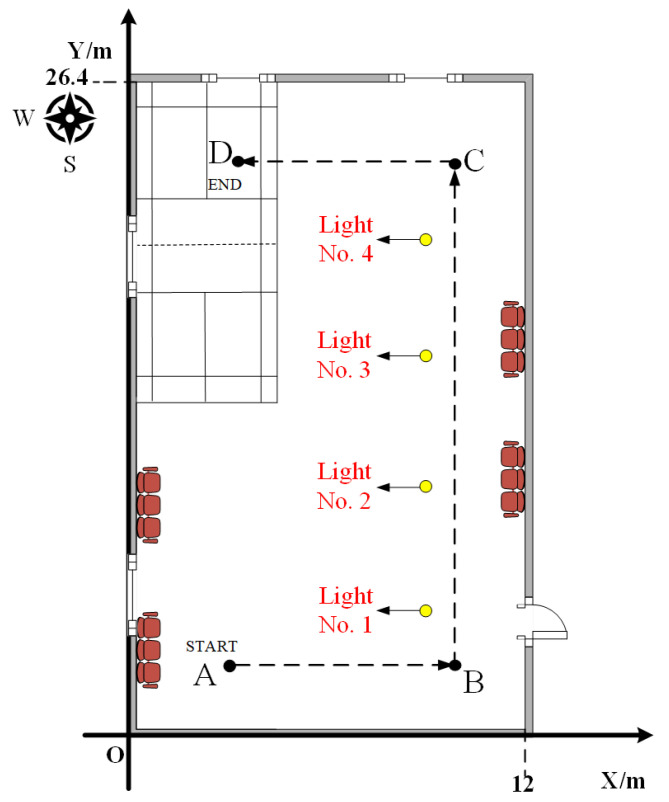
Schematic diagram of experimental scenario I.

**Figure 9 sensors-23-07929-f009:**
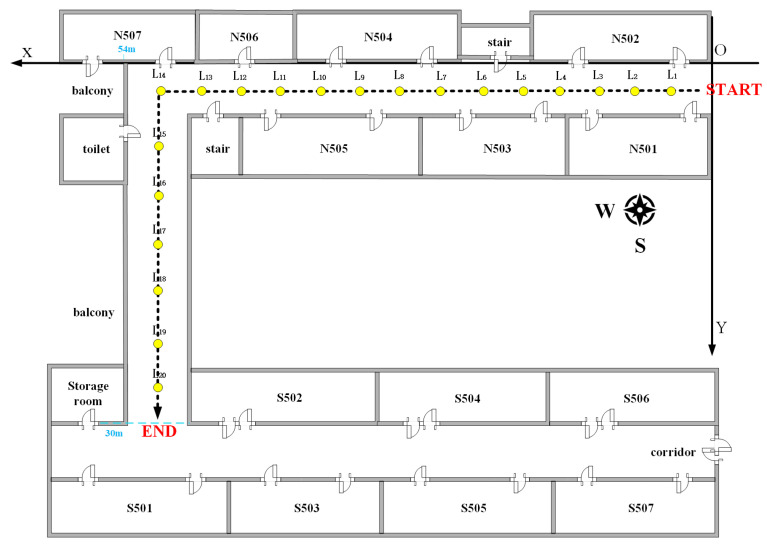
Schematic diagram of experimental scenario II.

**Figure 10 sensors-23-07929-f010:**
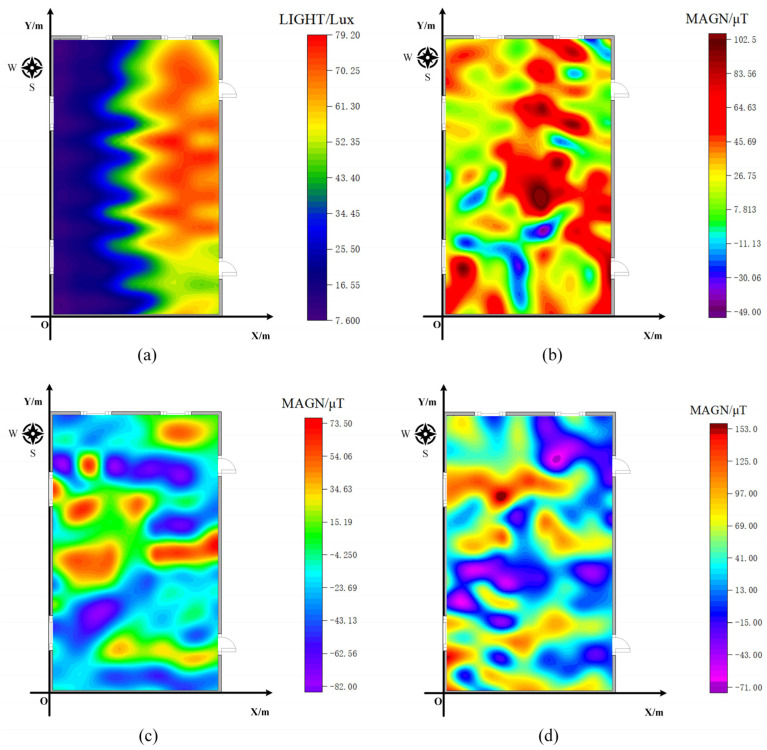
Geomagnetic and light intensity distribution of scenario I. (**a**) Light intensity distribution of scenario I; (**b**) Geomagnetic x-axis data distribution of scenario I; (**c**) Geomagnetic y-axis data distribution of scenario I; (**d**) Geomagnetic z-axis data distribution of scenario I.

**Figure 11 sensors-23-07929-f011:**
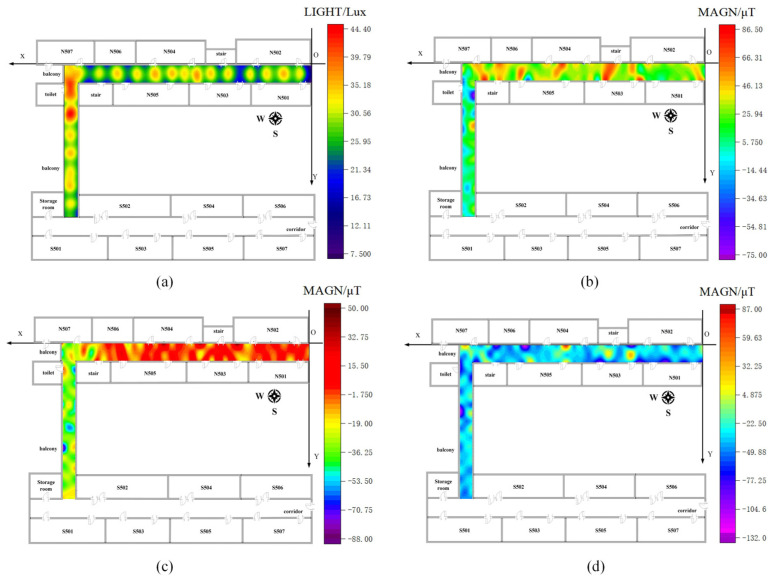
Geomagnetic and light intensity distribution of scenario II. (**a**) Light intensity distribution of scenario II; (**b**) Geomagnetic x-axis data distribution of scenario II; (**c**) Geomagnetic y-axis data distribution of scenario II; (**d**) Geomagnetic z-axis data distribution of scenario II.

**Figure 12 sensors-23-07929-f012:**
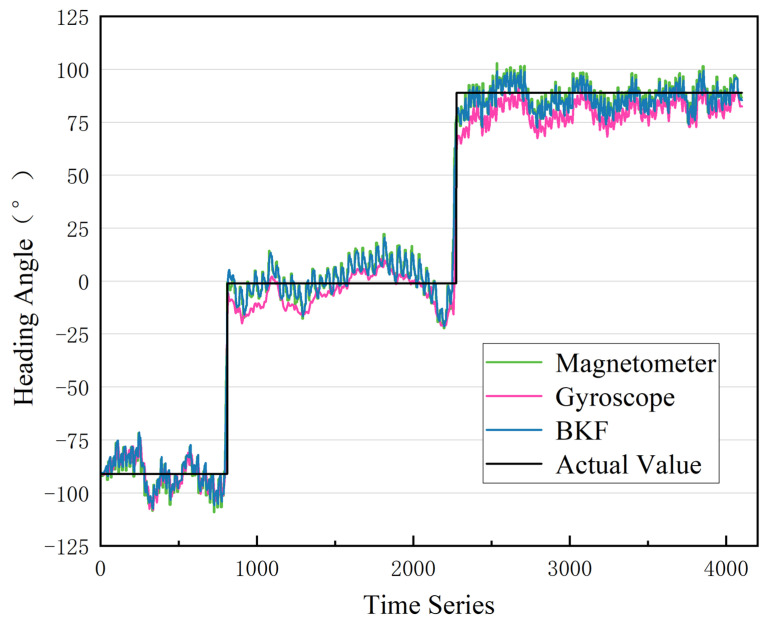
BKF fusion results.

**Figure 13 sensors-23-07929-f013:**
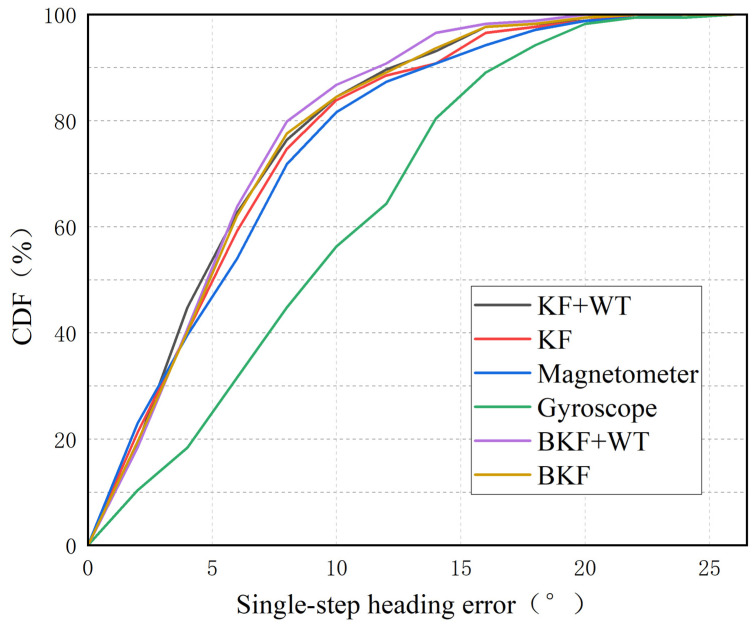
Cumulative distribution of single-step heading error.

**Figure 14 sensors-23-07929-f014:**
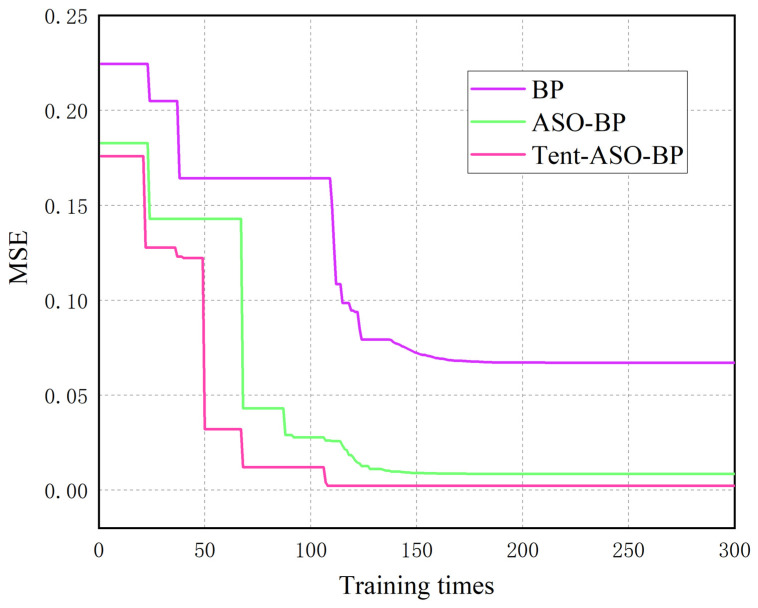
Comparison of convergence rates.

**Figure 15 sensors-23-07929-f015:**
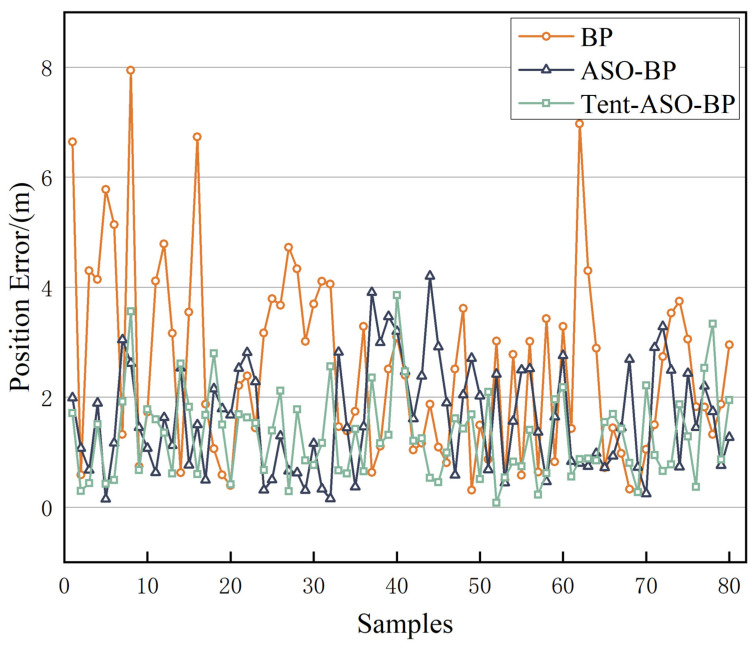
Positioning error statistics.

**Figure 16 sensors-23-07929-f016:**
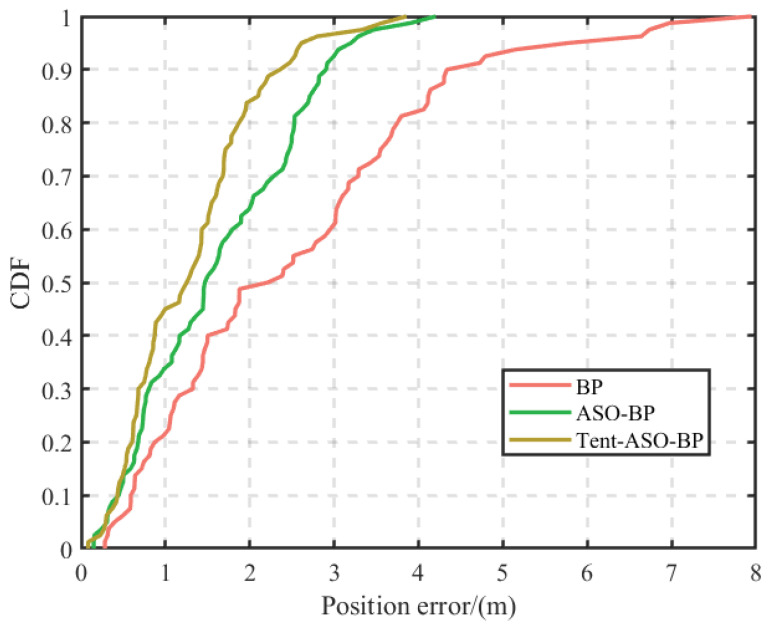
Cumulative distribution of positioning errors for different algorithms.

**Figure 17 sensors-23-07929-f017:**
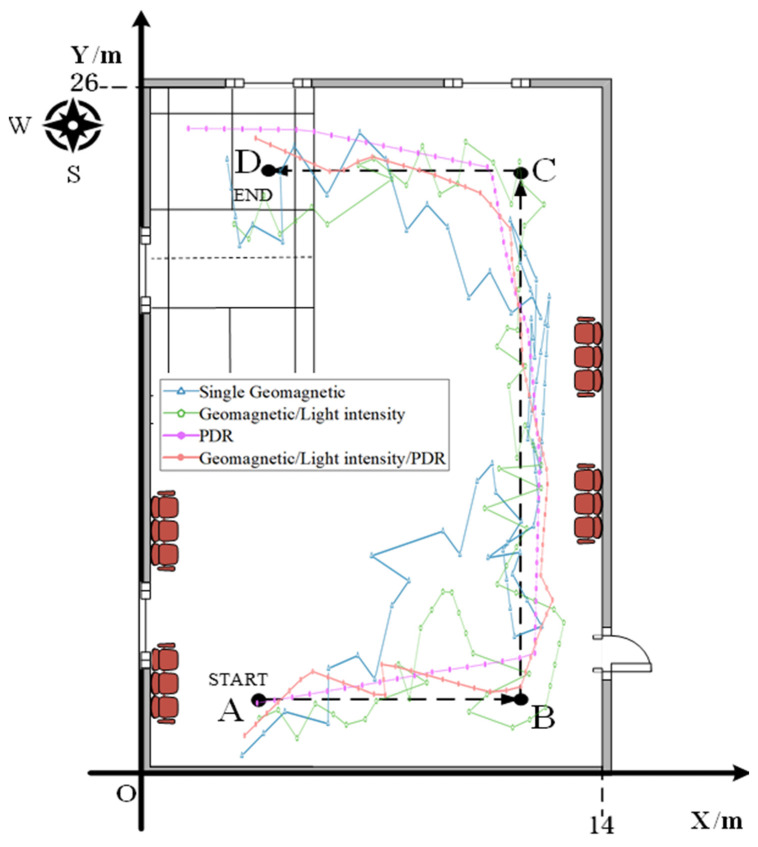
The trajectory positioning results of experimental scenario I.

**Figure 18 sensors-23-07929-f018:**
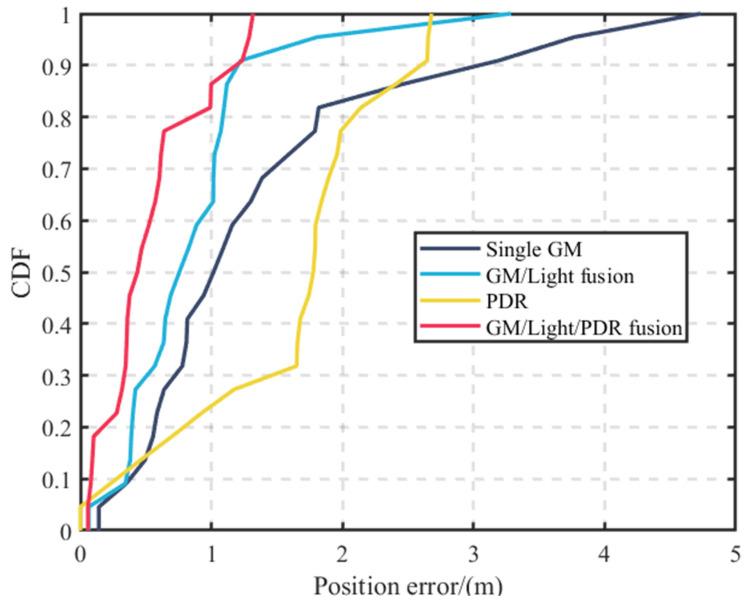
Cumulative distribution of positioning errors for scenario I.

**Figure 19 sensors-23-07929-f019:**
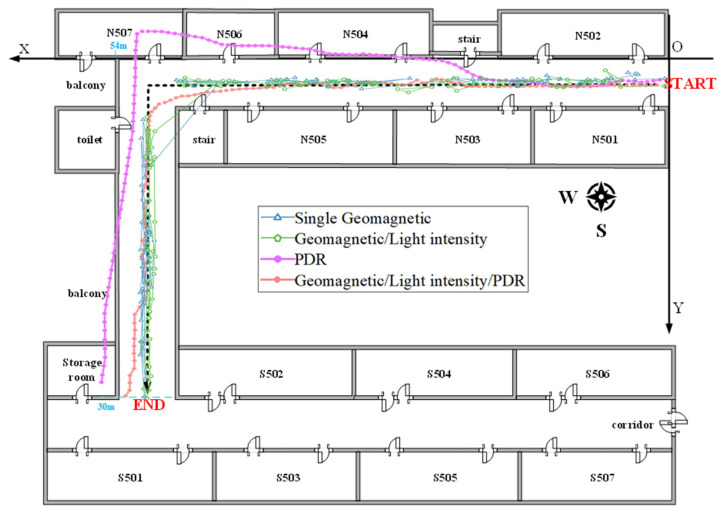
The trajectory positioning results of experimental scenario II.

**Figure 20 sensors-23-07929-f020:**
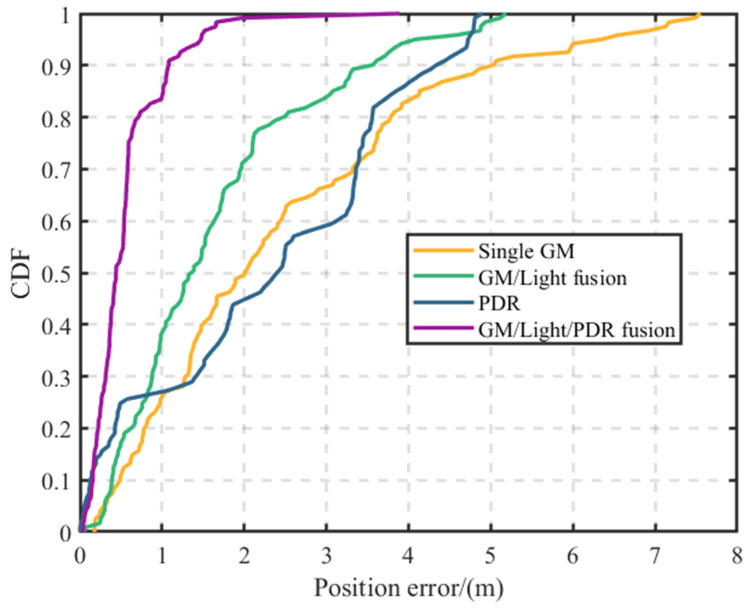
Cumulative distribution of positioning errors for scenario II.

**Table 1 sensors-23-07929-t001:** Single-step heading angle errors.

Property	WT Processing				No Processing			
	BKF	KF	Mag	Gyro	BKF	KF	Mag	Gyro
mean	5.45	5.76	6.30	7.71	5.63	5.97	6.21	9.21
variance	15.42	18.83	25.27	22.11	18.23	22.01	23.91	28.25
CDF = 30%	3.01	3.13	2.78	4.46	2.93	3.03	2.80	5.56
CDF = 60%	5.24	5.65	6.52	8.95	5.94	6.05	6.49	10.48
CDF = 90%	11.37	12.79	14.61	14.13	12.09	13.54	13.17	16.13

**Table 2 sensors-23-07929-t002:** Comparison of positioning experimental results.

Algorithm	Min Error/m	Max Error/m	Mean Error/m	SD/m
Tent-ASO-BP	0.08	3.85	1.31	0.80
ASO-BP	0.24	4.21	1.63	0.98
BP	0.31	7.97	2.51	1.72

**Table 3 sensors-23-07929-t003:** Comparison of positioning accuracy. Positioning accuracy comparison of scenario I.

Methods	Min/m	Max/m	Mean/m	SD/m
Single GM	0.14	3.95	1.52	1.13
GM/Light fusion	0.06	3.28	1.02	0.68
PDR	0.00	2.68	1.63	0.74
GM/Light/PDR fusion	0.05	1.31	0.56	0.41

**Table 4 sensors-23-07929-t004:** Comparison of positioning accuracy. Positioning accuracy comparison of scenario II.

Methods	Min/m	Max/m	Mean/m	SD/m
Single GM	0.18	7.56	2.45	1.82
GM/Light fusion	0.05	5.18	1.67	1.22
PDR	0.00	5.02	2.31	1.53
GM/Light/PDR fusion	0.04	1.88	0.75	0.45

## Data Availability

Not applicable.

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
