# Peer review of "A Fusion Positioning Method for Indoor Geomagnetic/Light Intensity/Pedestrian Dead Reckoning Based on Dual-Layer Tent–Atom Search Optimization–Back Propagation"

_sensors, 2023, doi:10.3390/s23187929_

Round 1

Reviewer 1 Report

REVISION MANUSCRIPT Sensors- 2563646:

“A Fusion Positioning Method for Indoor Geomagnetic/Light Intensity/PDR Based on Dual-Layer Tent-ASO-BP”

Editor

General comments:

The authors proposed a study based on geomagnetic indoor positioning. They suggested a fusion positioning method combining geomagnetic data, light intensity, and Pedestrian Dead Reckoning (PDR) based on hierarchical optimization. In addition, their approach employs the Atom Search Optimization (ASO) algorithm after chaos mapping to enhance the Back Propagation (BP) algorithm, resulting in the Tent-ASO-BP positioning model. Therefore, I found the research interesting. However, some suggestions and minor revisions need to be applied in order to consider for publication.

Specific comments:

Comment 1: In the introduction, it would be beneficial to include more background information from other studies related to the geomagnetic approach. This would provide context for the current study and highlight its novelty.

Comment 2: Regarding the position trajectory part, what is the reference system for the position coordinates? and how do you validate these positional results?

Comment 3: In Equation 2, the strategy they use to calculate d is very imprecise. And I think it is the variable that has the greatest difficulty to be measured, since, in order to calculate it with the method proposed in Section 2.1.2, the person must follow a linear trajectory. Also, the size of the length of a step is different in each person.

Comment 4: On line 197, at the point and followed, there is an extra T.

Comment 5: The errors shown in Figure 14 and discussed in lines 513-517 are very large, although it has been reduced by 50% with the proposed Ten-ASO-BP method.

Comment 6: In the experiments, how did the person keep the phone or sensors static while walking? Since any movement is detected by the sensors and the trajectory followed by the user is deviated.

Comment 7: All figures in this manuscript need to be embellished. Also improve their quality.

none

Reviewer 2 Report

The authors present an indoor positioning method combining geomagnetic, light intensity, and PDR data using a two-layer Tent-ASO-BP model. The topic is interesting and relevant, as indoor positioning remains an important challenge. The manuscript is overall well-written. However, there are some issues that should be addressed before acceptance:

Major points:

The introduction provides a good overview of existing work on indoor positioning using geomagnetic data, visible light, and PDR. However, the authors should more clearly explain the novel contributions of their proposed approach compared to prior art. How exactly does their two-layer model and integration of multiple data sources improve upon previous methods?

More details are needed on the Tent-ASO-BP model and algorithm. The current description in Section 3.4 is rather high-level. Are there any hyperparameters that need to be set? How were they chosen?

The authors claim their approach enhances accuracy and stability across different scenarios, but the experimental validation is limited to two sites. Testing on more diverse environments (e.g. different floor layouts, building materials, etc.) would strengthen the results. Discussion of computational complexity and scalability to larger spaces is also lacking. This should be added into discussion (or limitation) section.

Minor points:

Abstract could be condensed with more precise language.

Some figures are blurry and could be improved (e.g., Figs 10).

References need to be formatted consistently and a few appear incorrect.

Some minor grammar issues throughout, recommend proofreading thoroughly.

Minor editing of English language required
